# Essential Amino Acid Ingestion Facilitates Leucine Retention and Attenuates Myofibrillar Protein Breakdown following Bodyweight Resistance Exercise in Young Adults in a Home-Based Setting

**DOI:** 10.3390/nu14173532

**Published:** 2022-08-27

**Authors:** Marcus Waskiw-Ford, Nathan Hodson, Hugo J. W. Fung, Daniel W. D. West, Philip Apong, Raza Bashir, Daniel R. Moore

**Affiliations:** 1Faculty of Kinesiology and Physical Education, University of Toronto, Toronto, ON M5S 2C9, Canada; 2KITE Research, Toronto Rehabilitation Institute, Toronto, ON M5G 2A2, Canada; 3Iovate Health Sciences International, Toronto, ON L6M 2R7, Canada

**Keywords:** essential amino acids, protein metabolism, oxidation, resistance exercise, bodyweight, home-based, protein metabolism

## Abstract

Home-based resistance exercise (RE) has become increasingly prevalent, but its effects on protein metabolism are understudied. We tested the effect of an essential amino acid formulation (EAA+: 9 g EAAs, 3 g leucine) and branched-chain amino acids (BCAAs: 6 g BCAAs, 3 g leucine), relative to a carbohydrate (CHO) placebo, on exogenous leucine retention and myofibrillar protein breakdown following dynamic bodyweight RE in a home-based setting. Twelve recreationally active adults (nine male, three female) participated in a double-blind, placebo-controlled, crossover study with four trial conditions: (i) RE and EAA+ (EX-EAA+); (ii) RE and BCAAs (EX-BCAA); (iii) RE and CHO placebo (EX-CHO); and (iv) rest and CHO placebo (REST-CHO). Total exogenous leucine oxidation and retention (estimates of whole-body anabolism) and urinary 3-methylhistidine:creatinine ratio (3MH:Cr; estimate of muscle catabolism) were assessed over 5 h post-supplement. Total exogenous leucine oxidation and retention in EX-EAA+ and EX-BCAA did not significantly differ (*p* = 0.116) but were greater than EX-CHO (*p* < 0.01). There was a main effect of condition on urinary 3MH:Cr (*p* = 0.034), with post hoc analysis revealing a trend (*p* = 0.096) for reduced urinary 3MH:Cr with EX-EAA+ (32%) compared to EX-CHO. By direct comparison, urinary 3MH:Cr was significantly lower (23%) in EX-EAA+ than EX-BCAA (*p* = 0.026). In summary, the ingestion of EAA+ or BCAA provided leucine that was ~60% retained for protein synthesis following home-based bodyweight RE, but EAA+ most effectively attenuated myofibrillar protein breakdown.

## 1. Introduction

The preservation and growth of lean body mass is critical to athletic performance and healthy physiology [1,2,3,4]. The accretion of lean body mass can be facilitated by resistance exercise (RE) training, including training with low external loads but high volitional effort [5,6,7]. The protein component of lean body mass is in constant turnover through the reciprocal processes of protein synthesis and protein breakdown [8,9]. Protein turnover increases with RE [10], and over time, the net anabolic effect contributes to the growth of lean body mass. Recently, a surge in the prevalence of home-based exercise, which frequently utilizes bodyweight as resistance, has occurred in healthy adults [11]. However, the effects of bodyweight RE on protein metabolism and how the anabolic potential of this type of exercise is best supported nutritionally are notably understudied, particularly in home-based contexts where noninvasive approaches would be required.

Previous work involving the noninvasive assessment of protein metabolism has measured exogenous amino acid (AA) oxidation to determine protein requirements following RE in different contexts [12,13,14]. The approach used in many noninvasive studies involved constant feeding protocols and stable isotope tracers (e.g., 1-[^13^C]-phenylalanine) that may be primarily metabolized within the liver [15]. Although effective in estimating daily protein requirements, this method does not necessarily reflect the metabolism of a bolus dose of essential amino acids (EAAs), which may be more anabolic than smaller pulse feedings [16,17]. A leucine-based tracer may also be more relevant in estimating post-RE AA oxidation, as its first-pass extraction in splanchnic tissues is relatively low [18,19,20], and it is primarily oxidized in skeletal muscle [21]. Since exogenous leucine is either oxidized and excreted in the form of CO_2_ (and urea) or retained for protein synthesis [22], the retention of exogenous leucine following feeding can provide an estimate of whole-body protein synthesis [16,23]. Exogenous leucine retention can be determined noninvasively by feeding “labeled” AA sources and then measuring ^13^CO_2_ production over the subsequent 5–6 h postprandial period [16,20,24,25]. We have recently demonstrated that an isotopically labeled amino acid and carbohydrate beverage consumed after high-intensity whole-body resistance exercise results in greater [^13^C]-leucine retention than when consumed at rest [20], highlighting its utility in detecting increased whole-body anabolism over the subsequent 5–6 h postprandial period. However, considering that all EAAs are needed as substrates for protein synthesis, it is unclear if ingesting all EAAs facilitates greater exogenous leucine retention than only a subset of EAAs, such as the branched-chain amino acids (BCAAs) [26].

Although RE increases protein synthesis, in the absence of adequate EAA ingestion, protein breakdown must increase in parallel to supply the necessary AA substrates for synthesis [10], resulting in a net catabolic state. An AA released during the degradation of myofibrillar proteins is 3-methylhistidine (3MH). Since the post-translational methylation of histidine to 3MH prevents its reutilization following breakdown [27], its only fate is to be excreted in urine [28]. Urinary concentrations of 3MH may therefore serve as a practical noninvasive biomarker of myofibrillar protein breakdown [29]. The consumption of all EAAs has the capacity to attenuate the rise in post-RE urinary 3MH [30,31], presumably due to the fact that it directly provides the substrates needed for increased protein synthesis. However, because BCAAs only provide a subset of EAAs, their ability to mitigate post-RE protein breakdown is less certain [26].

The primary objective of this study was to investigate exogenous leucine oxidation/retention with EAA and BCAA ingestion in comparison to a carbohydrate (CHO) placebo following bodyweight RE in a home-based setting. A secondary objective was to determine whether bodyweight RE influences exogenous leucine oxidation/retention in the absence of AA intake as compared to a rested condition. We also sought to determine the effect of post-RE EAA and BCAA ingestion, relative to placebo, on urinary 3MH (normalized to urinary creatinine, 3MH:Cr) as an index of myofibrillar protein breakdown. To address these objectives, we conducted a double-blind, placebo-controlled, crossover, home-based study design with four trial conditions: (i) RE and an EAA formulation (EX-EAA+); (ii) RE and BCAAs (EX-BCAA); (iii) RE and placebo (EX-CHO); and (iv) rest and placebo (REST-CHO). We hypothesized that post-RE exogenous leucine retention would be greater with EAA+ or BCAAs compared to placebo, and only EAA+ would have a significant effect on attenuating urinary 3MH:Cr.

## 2. Materials and Methods

### 2.1. Participants

Twelve recreationally active and healthy young adults (9M, 3F) provided written informed consent to participate in a protocol that was written in accordance with standards set by the revised (2008) Declaration of Helsinki and approved by the research ethics board at the University of Toronto, Canada (protocol #27067). This study was pre-registered at https://www.clinicaltrials.gov (last accessed on 23 August 2022) as NCT04850820. 

Participants completed a Physical Activity Readiness Questionnaire [32] and the International Physical Activity Questionnaire [33] to verify that they could safely complete the RE protocol. Habitual diet data was collected over three days using self-reported data that participants manually inputted into MyFitnessPal (Under Armour, Baltimore, MD, USA). Participants were carefully instructed by study investigators to correctly record and input their habitual diet data.

### 2.2. Metabolic Trials

Trial order was randomized and counterbalanced. All study procedures took place in the participants’ place of residence. Participants were instructed to minimize activity (prolonged walking or standing) during trials, but were otherwise free to work or do other home activities. Structured physical activity was prohibited for at least 48 h prior to trials and caffeine was prohibited at least 12 h prior to trials. On the day before each trial, participants adhered to a standardized diet that was based on their respective habitual diet logs. Aside from the trial supplements, participants refrained from all food and beverages during the trial, but were permitted to consume water ad libitum.

An overview of the metabolic trial protocol can be found in Figure 1. At ~08h00 following an overnight 10 h fast, participants voided their bladders and performed a 40 min whole-body bout of RE as described below. Immediately following RE, participants collected baseline breath and spot urine samples. After consumption of the trial supplement (t = 0 min), participants collected breath samples every 20–30 min (at t = 20, 40, 60, 80, 100, 120, 140, 160, 180, 210, 240, 270, and 300 min) over a 5 h measurement period. During this 5 h period, all urine excreted by participants was collected and pooled in a large urine container with a final void of their bladder at the end of the period. After the final void, participants extracted a representative sample from the large urine container into a spot urine container that was immediately stored at −20 °C. The rested trial (REST-CHO) followed the same protocol but did not include RE.

### 2.3. Resistance Exercise Protocol

The 40 min RE protocol was identical for each exercise trial and consisted of six 5 min sets targeting different muscle groups in the following order: legs, chest, shoulder, back, abdominal, and quadriceps (Table 1). Each set involved five 1 min calisthenic and plyometric exercises (e.g., pushups, squat jumps) that were performed continuously with maximum effort. Participants were given a 2 min break between each set and were permitted to drink water during breaks. Exercise sessions were monitored via video call by a study investigator to ensure adherence and participant safety. Participants were asked to rate their rating of perceived exertion at the beginning and end of each exercise bout using the Borg CR10 scale [34].

### 2.4. Trial Supplements

All supplements were prepared in powder form and provided by Iovate Health Sciences International (Iovate Health Sciences International Inc., Oakville, ON, Canada) and flavour-matched to prevent unblinding. The EAA formulation (hereafter referred to as EAA+) used in the present study contained 9.07 g of free EAAs (3.00 g leucine, 2.17 g lysine, 1.67 g threonine, 750 mg isoleucine, 750 mg valine, 334 mg phenylalanine, 250 mg histidine, and 150 mg methionine). The EAA+ supplement was designed to enhance muscle growth and recovery during chronic resistance training and was originally intended to be tested in a resistance training study. However, due to the COVID-19 pandemic and restrictions to face-to-face research, our research was forced to adapt to a home-based study, as described herein. Nevertheless, we will outline the rationale for the composition of the EAA+ supplement designed to facilitate muscle growth and recovery. Specifically, EAA+ contained additional amino acids and natural botanical compounds, including: 1.20 g of citrulline, 805 mg of glutamine, 689 mg of *Schisandra chinensis*, 200 mg of *Lycium barbarum*, and 50.0 mg of ferulic acid. Citrulline has been reported to improve nitric oxide-mediated vasodilation [35], enhance muscle recovery following RE in men [36], and protect skeletal muscle cells from catabolic stimuli [37] whereas glutamine, which may influence protein synthesis by modulating mammalian target of rapamycin complex 1 (mTORC1) activity in cell culture [38], has been reported to enhance muscle recovery following RE in men [39]. The inclusion of the natural botanical compounds was based on preclinical evidence that *Schisandra chinensis* and compounds of the *Lycium* family have been reported to promote myoblast differentiation and mTORC1-mediated protein synthesis, as well as inhibit ubiquitin-proteasome-mediated protein degradation in rodent [40] and skeletal muscle cell models [41]. Lastly, ferulic acid is a notable anti-oxidant that is purported to support muscle growth during resistance training [42]. 

The BCAA supplement was designed to be leucine-matched and isocaloric to EAA+ and consisted of 6.00 g of free BCAAs (3.00 g leucine, 1.50 g isoleucine, 1.50 g valine) and 5.94 g of maltodextrin. The placebo supplement (CHO) was also designed to be isocaloric to EAA+ and contained 12.1 g of maltodextrin. All supplements were enriched with the addition of 100 mg of free 99% 1-[^13^C]-leucine (Cambridge Isotope Laboratories Inc., Tewksbury, MA, USA). The EAA+ and BCAA supplements therefore contained a total of 3.1 g of leucine and an enrichment of 3.3%. The only leucine in the CHO supplement was the 100 mg of 1-[^13^C]-leucine (enrichment of 99%). Participants prepared trial drinks by first adding 100 mL of water, mixing, and consuming, followed by adding a second 100 mL and mixing to ensure full consumption of the supplement.

### 2.5. Analysis of Breath Samples

Participants collected their own breath samples using provided equipment (Easy-Sampler, QuinTron Instrument Company, Milwaukee, WI, USA) into a sterile vacutainer. Breath samples were stored at room temperature before analysis of ^13^CO_2_ enrichment by continuous-flow isotope ratio mass spectrometry (ID-Microbreath; Compact Science Systems, Newcastle, UK).

### 2.6. Analysis of Urine Samples

Urine samples collected by participants were kept at −20 °C in plastic containers containing glacial acetic acid as a chemical preservative before being aliquoted and stored at −80 °C until analysis. The 3MH concentration was measuring by using a commercially available ELISA kit (cat. MBS7606614, MyBioSource, San Diego, CA, USA). Urinary 3MH measures were normalized to urinary creatinine (3MH:Cr; μmol/mmol) to increase precision and account for sample dilution [43,44]. Urinary creatinine concentrations were measured using a QuantiChrom Creatinine Assay Kit (cat. DICT-500, BioAssay Systems, Hayward, CA, USA).

### 2.7. Calculations

Exogenous leucine oxidation (Exo Ox) was calculated according to the following formula as previously described [16]:Exo Ox = (EiCO_2_/diet Leu Ei) × VCO_2_ × 1/k (1)
where EiCO_2_ is the enrichment of ^13^CO_2_ measured in breath samples (in atom percent excess; APE), diet Leu Ei is the leucine enrichment of the supplement that was ingested (in atom percent excess), VCO_2_ is the rate of carbon dioxide production (in μmol/kg/min), and k is a correction factor to account for the incomplete recovery of ^13^CO_2_ in breath samples [45]. EiCO_2_ at each time point was calculated as the difference between baseline enrichment and the enrichment of the breath sample at that time point. To limit the potential confounding effects of exercise on natural ^13^C baseline enrichment [46], baseline enrichment was calculated as the average enrichment of the pre-RE and immediate post-RE (t = 0) breath samples. A k of 0.8 and 0.7 was used for the AA and CHO conditions, respectively [47]. 

As this study was completely home-based, we were unable to measure VCO_2_. We therefore obtained an estimate of resting VCO_2_ [48], in μmol/kg/min, by multiplying body surface area (BSA) by 300 mmol CO_2_/h [49]. Previous work from our lab has indicated that this approach yields comparable data to resting VCO_2_ measured via indirect calorimetry [20]. BSA was estimated according to Haycock’s equation [50]:BSA = 0.024265 × height^0.3964^ × body mass^0.5378^(2)
where height is expressed in centimeters and body mass is expressed in kilograms. Because post-RE elevations in resting energy expenditure may last for up to 60 min [51], we increased the estimated resting VCO_2_ by 10% for the 20 min, 40 min, and 60 min time points, which is an increase that may be expected with the type of RE performed in this study [52]. 

Total trial exogenous leucine oxidation, expressed in μmol/kg, was obtained by calculating the area under the curve (AUC) of exogenous leucine oxidation over the 5 h measurement period. Total exogenous leucine retention (μmol/kg) was then calculated as the difference between total exogenous leucine oxidation and leucine ingestion. Exogenous leucine oxidation expressed in %/h was calculated as the fraction of ingested leucine oxidized at each time point, whereas total exogenous leucine oxidation expressed in % was calculated as the cumulative percentage of ingested leucine oxidized over the 5 h measurement period.

### 2.8. Statistical Analysis

The primary outcome for the present study was total exogenous leucine retention as previously done in our laboratory [20]. In our previous parallel-group design, the smallest difference between treatment groups in total exogenous leucine retention over 5 h was 9.4 μmol/kg with a common SD of 5.9μmol/kg. Based on this, and with an α = 0.05 and 1-β = 0.80, n = 6 was determined to be sufficient to detect a significant difference between EX-EAA+ and EX-CHO in the present study. To account for potential dropouts and increase power for secondary and tertiary outcomes, we recruited n = 12 participants.

Data were analyzed in GraphPad Prism (version 9.3.0, GraphPad Software, San Diego, CA, USA), with significance set at *p* < 0.05. All data were checked for violations of normality by Shapiro–Wilk test. Values identified as two standard deviations from the mean were classified as outliers and winsorized to the next less extreme value within the same condition [53]. Exogenous leucine oxidation for all conditions was analyzed by two-way repeated measures ANOVA (condition × time). Total exogenous leucine oxidation, total exogenous leucine retention, and urinary 3MH:Cr for all conditions were analyzed by one-way repeated measures ANOVA (condition). If a significant main effect was detected, a Holm–Sidak post hoc test and effect size (ES) analysis was conducted to compare EX-EAA+, EX-BCAA, and REST-CHO conditions to the EX-CHO control condition. To determine the potential effect of habitual dietary protein intake on any outcomes, all variables were also analyzed by ANCOVA with habitual dietary protein intake as the covariate. To identify potential differences between the AA conditions (EX-EAA+ and EX-BCAA), a secondary exploratory analysis was conducted by paired *t*-test for total exogenous leucine oxidation, total exogenous leucine retention, and urinary 3MH:Cr.

## 3. Results

Participant characteristics can be found Table 2. Mean post-RE ratings of perceived exertion were overall the equivalent of “very hard”, with no differences in mean (±SD) ratings between trial conditions (7.61 ± 1.39, 7.75 ± 1.46, and 7.85 ± 1.45 for the EX-EAA+, EX-BCAA, and EX-CHO conditions, respectively; *p* = 0.493).

### 3.1. Relative Exogenous Leucine Oxidation

Exogenous leucine oxidation (expressed in %/hr; Figure 2A) increased during the 5 h measurement period (main effect of time, *p* < 0.01) with an effect of condition (*p* < 0.01) and condition × time interaction (*p* < 0.01). A main effect of condition was found for total exogenous leucine oxidation (expressed in %; Figure 2B; *p* < 0.01), with EX-EAA+ (*p* < 0.01, ES = 2.84), EX-BCAA (*p* < 0.01, ES = 3.22), and REST-CHO (*p* < 0.01, ES = −1.01) all significantly differing from EX-CHO. Total exogenous leucine oxidation (%) with EX-EAA+ was not significantly different than EX-BCAA (*p* = 0.147, ES = 0.45). Habitual protein intake had no effect on any parameters when included as a covariate during analysis.

### 3.2. Absolute Exogenous Leucine Retention

A main effect of condition was found for total exogenous leucine retention (expressed in μmol/kg; Figure 2C; *p* < 0.01), with EX-EAA+ (*p* < 0.01, ES = 7.75), EX-BCAA (*p* < 0.01, ES = 9.59), and REST-CHO (*p* < 0.01, ES = 0.94) all significantly greater than EX-CHO. No significant differences were found between EX-EAA+ and EX-BCAA for this outcome (*p* = 0.116, ES = 0.49). Habitual protein intake had no effect on any parameters when included as a covariate during analysis.

### 3.3. Urinary 3-Methylhistidine:Creatinine

A sensitivity analysis was performed as a pivotal outlier was identified. Without winsorization, the data violated assumptions of normality, and no effect of condition was observed (*p* = 0.600). With outliers winsorized to the next less extreme value, all conditions passed assumptions of normality and a main effect of condition was observed (Figure 3, *p* = 0.034). Mean concentrations of 3MH:Cr, as compared to EX-CHO, were 32% (*p* = 0.096, ES = −0.70), 12% (*p* = 0.303, ES = −0.36), and 14% (*p* = 0.303, ES = −0.43) lower for EX-EAA+, EX-BCAA, and REST-CHO, respectively, with no post hoc comparison reaching statistical significance. However, 3MH:Cr was significantly lower (23%) with EX-EAA+ when compared to EX-BCAA (*p* = 0.026, ES = −0.74) by paired *t*-test.

## 4. Discussion

The main finding of this study was that an EAA formulation (EAA+) and leucine-matched BCAAs similarly increased total exogenous leucine oxidation and retention relative to a CHO control condition following bodyweight RE in a home-based setting. We also report that in the absence of AA intake, bodyweight RE decreased exogenous leucine retention compared to a rested condition. Finally, EAA+ ingestion generally attenuated the post-RE rise in urinary 3MH:Cr concentrations as compared to BCAA and CHO. 

It has been well-established that AA ingestion is accompanied by increased leucine oxidation [16,19,23,54], as the fraction of dietary leucine unable to be retained as substrate for protein synthesis is subsequently oxidized to CO_2_ [22]. Therefore, expectedly, the high dose of leucine in the EAA+ and BCAA supplements (3.1 g) resulted in increased exogenous leucine oxidation in our healthy young adults, with almost all oxidation occurring during the 5 h measurement period (Figure 2A). This aligns with previous studies that observed a rise in exogenous AA oxidation that was generally complete within ~5–6 h [16,24,25,48,55,56,57]. In comparison to these studies, peak exogenous leucine oxidation was greater (~15%/h) and occurred earlier (~60 min) in the present study, likely due to the ingestion of free AAs [24,25,58] as opposed to intact whole protein (e.g., milk or egg) previously used [16,24,25,48,55,56,57]. Interestingly, while habitual protein intake may affect AA oxidation during exercise [59], habitual protein intake had no effect on post-RE exogenous leucine oxidation when included as a covariate during analysis in the present study. This is consistent with previous work that did not observe any differences in exogenous AA oxidation with changes in dietary protein intake [56] and the suggestion that dietary amino acids are preferentially utilized for protein synthesis [16].

Previous research has shown that the retained portion of dietary leucine is mainly directed toward nonoxidative leucine disposal (i.e., NOLD), a proxy for protein synthesis [16], that is stimulated in part by the dose of leucine consumed [60]. This was corroborated by recent work from our laboratory, where it was found that the postprandial retention of exogenous leucine following RE generally correlated with myofibrillar protein synthesis rates and was greater than at rest [20]. This suggests that the retention of exogenous leucine in the present study, which was significantly greater with EAA+ and BCAA ingestion as compared to placebo (Figure 2C), was indicative of enhanced whole-body protein synthesis and may in part be reflective of skeletal muscle protein synthesis as well. It is important to note that our participants did consume 100 mg of 1-[^13^C]-leucine in the CHO conditions, but this trivial quantity of labeled leucine (equivalent to ~11 μmol/kg in an average 70 kg participant) has been commonly used via intravenous infusion to assess protein metabolism and does not affect whole-body or muscle protein metabolism [16,23,61]. 

To stimulate anabolism at home without equipment, our participants performed a high volume of low-load repetitions using only bodyweight as resistance (i.e., calisthenics and plyometrics) with limited breaks. Despite the use of low loads, substantial effort and muscular exertion can enhance protein synthesis [62] and stimulate training-induced increases in lean body mass [5,6,7]. Thus, bodyweight RE may facilitate greater *retention* of exogenous AA to assist with the remodeling and growth of skeletal muscle [9,63]. In contrast, the percentage of exogenous leucine that was retained in the present study was significantly lower (~3.7%) following RE than at rest (Figure 2C). While this study was not designed to elucidate the underlying mechanisms for this discrepancy given its home-based design, a potential explanation may lie in the degree of exercise intensity. With average Borg CR10 ratings of perceived exertion of ~8, the dynamic and continuous movements involved in our bodyweight RE protocol were performed at a high-intensity that presumably led to a significant increase in heart rate and oxygen consumption [52,64]. This substantial aerobic component could have elicited greater *endogenous* AA oxidation during exercise [9,19,65,66] relative to rest. Consequently, a change in endogenous AA concentrations post-RE [67] may have created an imbalance of AA substrates for protein synthesis and thus a larger proportion of exogenous leucine could not be retained and was oxidized. Our findings may agree with recent work from Reckman et al. [48] who found healthy young males oxidized significantly more of a 30 g bolus of labeled whey protein following vigorous aerobic exercise (60% VO_2_max) as compared to rest and lower intensity (30% VO_2_max) exercise [48]. Collectively, these data could suggest the existence of an exercise intensity threshold, which when surpassed (as it potentially was in the present study) results in significant AA oxidation during exercise and potentially during post-exercise recovery. Accordingly, high-intensity aerobic exercise in the postabsorptive state may dampen protein synthesis in the initial hours post-exercise [59]. Furthermore, respiratory exchange ratio has also been reported to decrease during recovery from exercise at the same time that energy expenditure (i.e., oxygen consumption) is increased [51]. Thus, it is unclear if our predicted VCO_2_ was indeed elevated after our RE and, subsequently, leucine retention was lower after exercise than at rest. Regardless, the substantially lower leucine retention in the absence of BCAA or EAA+ ingestion highlight the importance of adequate post-exercise dietary amino acid intake for individuals performing high-intensity RE to enhance its anabolic effect.

We found slightly greater exogenous leucine retention in EX-EAA+ when directly compared to EX-BCAA (moderate effect size), although this was not statistically significant (Figure 2C). This study was not powered to detect a difference between these conditions, and thus it remains unclear if these supplements facilitate equivalent exogenous leucine retention. It is possible that the provision of all EAAs results in a better balance of AAs for protein synthesis, and thus additional leucine was retained with EAA+. Alternatively, the equivalent leucine content with EAA+ and BCAA could have elicited a similar stimulation of protein synthesis [60], and by extension, a similar retention of leucine as substrate. We have also previously shown that BCAAs may be the main rate-limiting AAs [68] and that nonessential amino acids are not required for protein synthesis following aerobic exercise [69]. While this was found in a different exercise modality than the bodyweight RE of the present study, it remains possible that BCAA ingestion provided a sufficient proportion of the deficient EAA precursors for protein synthesis in the acute post-RE period and thus resulted in a comparable degree of exogenous leucine retention as EAA+.

If RE is performed without the intake of sufficient EAAs, then the resultant increase in protein synthesis must be accommodated by an increase in protein breakdown to supply the necessary AA substrates [10]. However, the adequate provision of exogenous EAAs mitigates the need to catabolize muscle tissue to sustain increases in protein synthesis [70,71]. Accordingly, we found a main effect of condition on urinary 3MH:Cr (Figure 3), although we may have been underpowered to elucidate specific differences between conditions in our post hoc analysis. Regardless, the ~32% lower urinary 3MH:Cr with EX-EAA+ as compared to EX-CHO is consistent with previous work. For instance, Bird et al. reported in multiple studies [30,31] that ~6 g of EAA supplementation mitigated the post-RE increase in urinary 3MH:Cr. This is further supported by our exploratory analysis that revealed EAA+ was significantly more effective (23%) at attenuating post-RE urinary 3MH:Cr than BCAAs. These findings are consistent with the notion that BCAA are less capable of reducing muscle protein breakdown as they do not provide all of the EAA substrates needed for muscle protein synthesis [26]. 

It should also be reiterated that the EAA+ formulation contained additional bioactive compounds (citrulline, glutamine, ferulic acid, and extracts from *Schisandra chinensis* and *Lycium barbarum*). If these compounds were to elicit any effects in the acute (5 h) post-RE period, it may have been most evident in our measurements of urinary 3MH:Cr. For example, anti-catabolic effects have been independently reported with citrulline (in vitro) [72] and glutamine (in clinical populations) [73], whereas the reported anti-inflammatory and/or anti-oxidant effects of ferulic acid [74], *Schisandra chinensis* [75], and *Lycium barbarum* [76] theoretically may have influenced protein breakdown. Although it cannot be confirmed that the additional compounds in EAA+ had no effects on study outcomes, these effects may be more prominent in a chronic training paradigm as opposed to our acute model. While our data suggest that the EAA+ and BCAA supplements had similar effects on whole-body protein synthesis, both BCAAs and CHO were notably less impactful at attenuating myofibrillar protein breakdown. Taken together and assuming any potential effects of the additional compounds included in EAA+ in this acute study were negligible, these results could suggest that a full complement of EAAs is necessary to maximize the anabolic effect (i.e., increased leucine retention and attenuated myofibrillar catabolism) of RE. 

The present study was unique in that, to our knowledge, it was the first study to evaluate post-RE protein metabolism entirely in a home-based setting. We show that the study design approach presented here is feasible and can provide insight on post-exercise protein metabolism outside of the laboratory. Moreover, as the prevalence of home or office-based exercise continues to rise [11], it becomes increasingly important to understand the metabolic consequences of bodyweight RE (i.e., high volume of low loads) that is commonly performed in these settings. In summary, our findings indicate that the high-intensity characteristics inherent to this type of exercise elevates exogenous leucine oxidation and therefore must be supported with adequate AA intake to promote the maintenance or growth of lean body mass. Both the EAA formulation (EAA+) and BCAAs tested in this study provided leucine that was ~60% retained for protein synthesis, but only the EAA+ supplement had a substantial effect on attenuating myofibrillar protein breakdown following RE. Collectively, our results suggest the consumption of all EAAs is necessary to optimize post-exercise anabolism.

## Figures and Tables

**Figure 1 nutrients-14-03532-f001:**
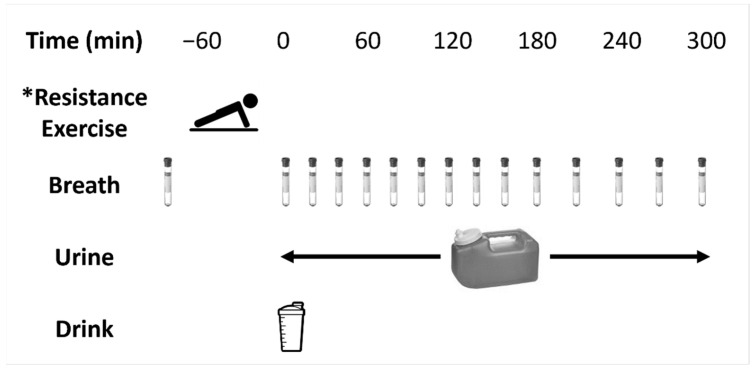
Overview of the metabolic trials. * The rested-carbohydrate (REST-CHO) trial followed the same protocol but did not include resistance exercise.

**Figure 2 nutrients-14-03532-f002:**
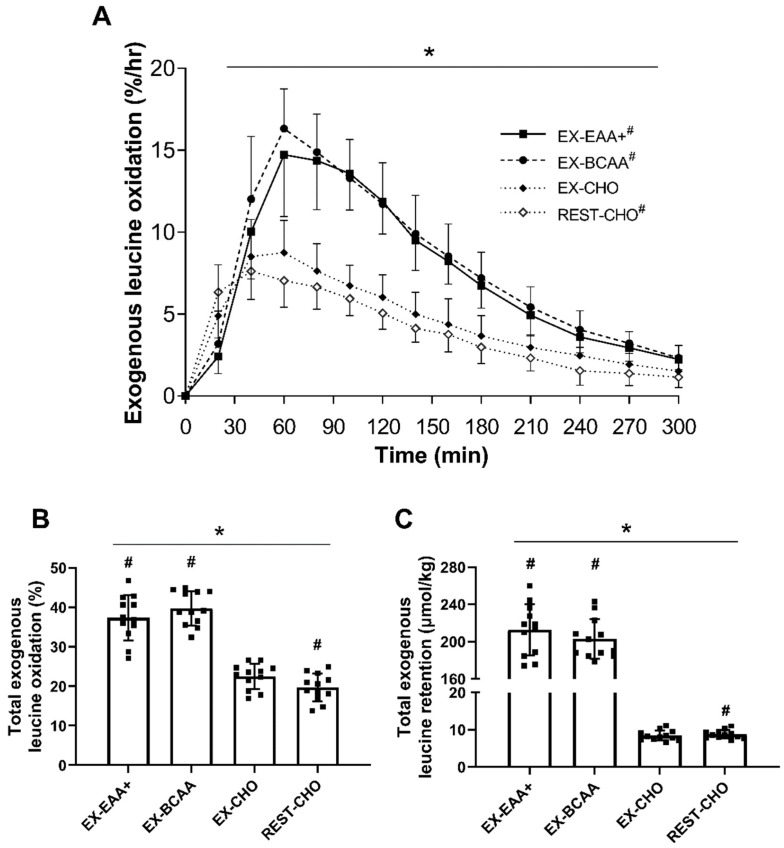
(**A**) Mean (±SD) exogenous leucine oxidation (%/hr). The conditions presented are as follows: EX-EAA+ (resistance exercise + essential amino acid formulation), solid line and closed squares (■); EX-BCAA (resistance exercise + branched chain amino acids), dashed line and closed circles (●); EX-CHO (resistance exercise + carbohydrate), dotted line and closed diamonds (♦); REST-CHO (rest + carbohydrate), dotted line and open diamonds (◊). * Main effect of time, condition, and condition × time interaction (*p* < 0.01). ^#^ Significantly different than the EX-CHO condition as determined by Holm–Sidak post hoc test (*p* < 0.01). (**B**) Mean (±SD) total exogenous leucine oxidation (%). Individual data points are represented by the closed squares (■). * Main effect of condition (*p* < 0.01). ^#^ Significantly different than the EX-CHO condition as determined by Holm–Sidak post hoc test (*p* < 0.01). (**C**) Mean (±SD) total exogenous leucine retention (μmol/kg). Individual data points are represented by the closed squares (■). * Main effect of condition (*p* < 0.01). ^#^ Significantly different than the EX-CHO condition as determined by Holm–Sidak post hoc test (*p* < 0.01).

**Figure 3 nutrients-14-03532-f003:**
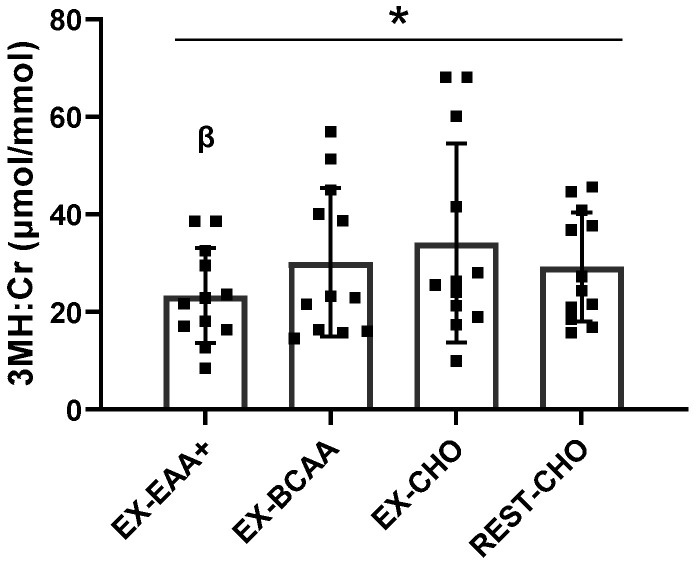
Mean (±SD) urinary 3-methylhistidine:creatinine (3MH:Cr) concentrations over the 5 h collection period following resistance exercise. Individual data points are represented by the closed squares (■). * Main effect of condition (*p* = 0.034). ^β^ Significant difference between EX-EAA+ (resistance exercise + essential amino acid formulation) and EX-BCAA (resistance exercise + branched chain amino acids) as determined by paired *t*-test (*p* = 0.026). EX-CHO, resistance exercise + carbohydrate; REST-CHO, rest + carbohydrate.

**Table 1 nutrients-14-03532-t001:** Outline of the bodyweight resistance exercise protocol. All exercises were performed maximally and continuously, and sets were separated by 2 min breaks.

Set	Exercise	Duration
Legs	Close stance squats	1 min
	Leaning Romanian deadlifts	1 min
	Reverse lunges with kickback	1 min
	Squat with calf jump	1 min
	Lunge pulses	1 min
Chest	Side-to-side push-ups	1 min
	Decline push-ups	1 min
	Reverse grip push-ups	1 min
	Push-up holds	1 min
	Pectoral crushers	1 min
Back	Reverse snow angels	1 min
	Seal push-ups	1 min
	Pulse rows	1 min
	Forward reaches	1 min
	Supermans	1 min
Shoulders	Front twist raises	1 min
	Switch side pulses	1 min
	Front air drivers	1 min
	Reverse Arnolds	1 min
	Bent triplexes	1 min
Abdominals	Rising flutters	30 s
	Abdominal rockers	30 s
	Toe touchers	30 s
	Penguin ankle taps	30 s
	Legs extended crunches	30 s
	Touch and go crunches	30 s
	Hip thrusts	30 s
	Bicycle crunches	30 s
	Leg crunches	30 s
	Buster crunches	30 s
Quadriceps	Explosive squat jumps	1 min
	Close to wide squat jumps	1 min
	Speed squats	1 min
	Squat pulses	1 min
	Squat hold	1 min

**Table 2 nutrients-14-03532-t002:** Participant characteristics.

Variable	Mean ± SD
Age (years)	26.92 ± 3.34
Estimated height (cm)	172.42 ± 8.37
Estimated weight (kg)	71.00 ± 9.06
Body mass index (kg/m^2^)	23.81 ± 2.40
Habitual dietary protein (g/kg/day)	1.76 ± 0.65
Habitual dietary energy (kcal/day)	2170.75 ± 599.70

## Data Availability

Not applicable.

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
