# Peer review of "Essential Amino Acid Ingestion Facilitates Leucine Retention and Attenuates Myofibrillar Protein Breakdown following Bodyweight Resistance Exercise in Young Adults in a Home-Based Setting"

_nutrients, 2022, doi:10.3390/nu14173532_

Round 1

Reviewer 1 Report

The manuscript is nice and is beautifully written. 

A major concern is the assumption of VCO2. Most of the parameters discussed, protein retention and excretion is based on the formula, which in turn used VCO2. Since home based study, (+pandemic), VCO2 could not be measures and understandable. But the assumption of VCO2 does not seems right  based on the references cited. please provide an explanation. 

Author Response

The manuscript is nice and is beautifully written. 

We thank the reviewer for taking the time to carefully review our manuscript and for the positive response. We have addressed their major concern below and made amendments to the manuscript to further strengthen our study rationale, in light of their helpful comments.

A major concern is the assumption of VCO2. Most of the parameters discussed, protein retention and excretion is based on the formula, which in turn used VCO2. Since home based study, (+pandemic), VCO2 could not be measures and understandable. But the assumption of VCO2 does not seems right  based on the references cited. please provide an explanation. 

Thank you for the comment, as the correct calculation of exogenous leucine oxidation is indeed important to our study findings. The calculations used to estimate resting VCO2 in our study (e.g., 300 mmol CO2/hr multiplied by body surface area) are correct and have previously been used in a similar study by other investigators (PMID: 31765432). In fact, this approach is commonly used in studies involving 13C breath tests to estimate resting VCO2 (PMID: 20443742).

Additionally, recent published work from our lab (PMID: 35609328) indicates that resting and post-exercise VCO2 measured by indirect calorimetry and estimated by the aforementioned calculations yields similar findings. This demonstrates that changes in breath enrichment are the primary determinant of the metabolic fate of the ingested tracer and that indirect calorimetry is not essential for estimating exogenous leucine oxidation; rather, an estimation of resting VCO2, as described in the Methods section of our manuscript, is sufficient (we have now included a sentence in the Methods section to state this in lines 204-205).

Reviewer 2 Report

The authors attempted to reveal the effect of ingested essential amino acids (EAA) or branched-chain amino acids (BCAAs) on exogenous leucine oxidation/retention and 3-methylhistidine (3MH), as an index of myofibrillar protein degradation, following home-based bodyweight resistance exercise. In addition, the authors sought to determine whether bodyweight resistance exercise influences exogenous leucine oxidation/retention in the absence of amino acid intake. The authors reported that ingested EAA formulation and BCAAs following resistance exercise increased exogenous leucine oxidation and retention, and decreased the 3MH/Cr. In the absence of amino acid intake (except isotope leucine), the authors demonstrated that resistance exercise increased exogenous leucine oxidation and decreased exogenous leucine retention. However, this reviewer has questions and thinks the conditions are inadequate.

Major comment

1)                  The exogenous leucine retention was calculated as the difference between total exogenous leucine oxidation and leucine ingestion. Why did EAA+ and BCAA increase the exogenous leucine retention, although the exogenous leucine oxidation in EAA+ and BCAA groups increased? Its calculation is correct?

2)                  The authors attempted to reveal the effect of ingesting EAAs on exogenous leucine retention as protein synthesis because protein synthesis needs all EAAs as substrate. However, the EAA+ has not only EAAs but other compounds affecting muscle protein metabolism. Thus, it is unclear that EAA in this study caused the results in EAA+.

3)                  The authors used resistance exercise and carbohydrate group (RE+CHO) as control, and compared the effect of EAA+ and BCAA to RE+CHO. However, the CHO did not have nitrogen comparable to EAA+ and BCAA supplements. Nitrogen is critical for amino acid biosynthesis; therefore, it is needed to meet the same nitrogen content as possible.

4)                  Due to the COVID-19 pandemic and restrictions to face-to-face research, the authors calculated the exogenous leucine oxidation using an estimated resting VCO2 in described in reference [48]. It is needed to indicate that the method using an estimated resting VCO2 is correct and appropriate.

Minor comment

1)                  Lines 97: habitual diet data was collected over three days.” Please, write the detailed collection protocol. Did the authors use a questionnaire or directly question the menu of habitual diet for three days?

2)                  Lines 134: tryptophan was excluded for regulatory reasons.”  What are the regulatory reasons? Please, write the reasons.

Author Response

The authors attempted to reveal the effect of ingested essential amino acids (EAA) or branched-chain amino acids (BCAAs) on exogenous leucine oxidation/retention and 3-methylhistidine (3MH), as an index of myofibrillar protein degradation, following home-based bodyweight resistance exercise. In addition, the authors sought to determine whether bodyweight resistance exercise influences exogenous leucine oxidation/retention in the absence of amino acid intake. The authors reported that ingested EAA formulation and BCAAs following resistance exercise increased exogenous leucine oxidation and retention, and decreased the 3MH/Cr. In the absence of amino acid intake (except isotope leucine), the authors demonstrated that resistance exercise increased exogenous leucine oxidation and decreased exogenous leucine retention. However, this reviewer has questions and thinks the conditions are inadequate.

We thank the reviewer for taking the time to carefully review our manuscript and for their helpful feedback. We have addressed their points below in detail and made revisions to the manuscript, in light of their helpful feedback.

Major comment

1)                  The exogenous leucine retention was calculated as the difference between total exogenous leucine oxidation and leucine ingestion. Why did EAA+ and BCAA increase the exogenous leucine retention, although the exogenous leucine oxidation in EAA+ and BCAA groups increased? Its calculation is correct?

Thank you for pointing this out as it may seem counterintuitive. However, the calculations are correct as it is well-documented (via intravenous and oral leucine tracers) that exogenous leucine oxidation increases with the consumption of leucine (PMID: 8997229) and that with foods or beverages higher in leucine elicited greater rates of leucine oxidation (PMID: 19056590). This would be consistent with EAA+ and BCAA increasing exogenous leucine oxidation compared to CHO in the present study.

                          The fraction of exogenous leucine oxidized is commonly around 30-40% (PMID: 8997229), and the remaining 60-70% of exogenous leucine is retained. Therefore, a proportion of the leucine in the EAA+ and BCAA supplements was retained in our study (~200μmol/kg, as shown in Figure 2C). This means that the absolute amount of exogenous leucine retained was greater with EAA+ and BCAA than CHO (which contained only trace amounts of isotopic leucine). Essentially, relative to CHO, both exogenous leucine oxidation and retention was greater with EAA+ and BCAA.

2)                  The authors attempted to reveal the effect of ingesting EAAs on exogenous leucine retention as protein synthesis because protein synthesis needs all EAAs as substrate. However, the EAA+ has not only EAAs but other compounds affecting muscle protein metabolism. Thus, it is unclear that EAA in this study caused the results in EAA+.

                          The reviewer is correct in raising this point. We are aware that the presence of glutamine, citrulline, and botanical compounds in the EAA+ supplement may have influenced our findings beyond any effects of the EAAs alone. We discussed the potential role of these compounds in both the Methods and Discussion sections, and stated that these compounds are unlikely to have acute (5h) effects based on preclinical data. This suggests that our findings are likely due to the EAAs in the EAA+ supplement. Additionally, our findings are consistent with studies using supplements only containing EAAs and other amino acids.

Nevertheless, we have now included additional text in the Discussion section to further state that we cannot confirm that these additional compounds in EAA+ did not impact our findings (lines 391-392).

3)                  The authors used resistance exercise and carbohydrate group (RE+CHO) as control, and compared the effect of EAA+ and BCAA to RE+CHO. However, the CHO did not have nitrogen comparable to EAA+ and BCAA supplements. Nitrogen is critical for amino acid biosynthesis; therefore, it is needed to meet the same nitrogen content as possible.

This is a great point, as many studies investigating amino acid supplements use a non-essential amino acid (NEAA) supplement as an isocaloric control. However, use of an isocaloric CHO control is not without precedence (PMID: 32349353). Both isocaloric NEAA and CHO controls are effective as they eliminate the concern of calories influencing outcomes and do not contain the key compound(s) under investigation (i.e., EAAs/leucine). To our knowledge, there is a lack of research indicating that the use of a NEAA control would impact the outcomes in our study differently than CHO. Nevertheless, our goal was to test the efficacy of essential amino acids in supporting whole body anabolism and made the decision to provide a nominal energy-matched control rather than just water. Regardless, we do not believe our results would be different if we had used a NEAA control condition as they have long been established to have no discernable effect on muscle (PMID: 10198297, 12217881) or whole body protein anabolism (PMID: 30145710).

4)                  Due to the COVID-19 pandemic and restrictions to face-to-face research, the authors calculated the exogenous leucine oxidation using an estimated resting VCO2 in described in reference [48]. It is needed to indicate that the method using an estimated resting VCO2 is correct and appropriate.

Thank you for the comment, as the correct calculation of exogenous leucine oxidation is indeed important to our study findings. The calculations used to estimate resting VCO2 in our study (e.g., 300 mmol CO2/hr multiplied by body surface area) are correct and have been used previously in a similar study (PMID: 31765432). In fact, this approach is commonly used in studies involving 13C breath tests to estimate resting VCO2 (PMID: 20443742).

Additionally, recent published work from our lab (PMID: 35609328) indicates that resting VCO2 measured by indirect calorimetry and estimated by the aforementioned calculations yields similar findings. This demonstrates that indirect calorimetry is not essential for estimating exogenous leucine oxidation and an estimation of resting VCO2, as described in the Methods section of our manuscript, is sufficient (we have now included a sentence in the Methods section to state this in lines 204-205).

Minor comment

1)                  Lines 97: “habitual diet data was collected over three days.” Please, write the detailed collection protocol. Did the authors use a questionnaire or directly question the menu of habitual diet for three days?

            Thank you for bringing this to our attention, as it may be unclear to readers. Habitual dietary data was collected using the software MyFitnessPal. We have now included additional information in the manuscript to clarify this (lines 97-100).

2)                  Lines 134: “tryptophan was excluded for regulatory reasons.”  What are the regulatory reasons? Please, write the reasons.

We can appreciate how this statement may have caused some confusion for the reviewer. From a regulatory standpoint, L-tryptophan was subject to an FDA recall and a ban that was in effect from 1991-2005 due to serious health concerns. FDA has since lifted the ban as it was concluded the ingredient in question was likely contaminated. However, the industry partner is cognizant that there may be residual apprehension in using the ingredient and some customers would not accepts products that contained it for a while after the ban. As this statement does not impact the scientific approach and is more from a consumer understanding standpoint, we have omitted it from the manuscript and have just left reference to the essential amino acids contained within the supplement.

Round 2

Reviewer 1 Report

I am sorry but I am not trying to harass the authors. Just wanted to make sure that the correct message is conveyed. 

The authors have assumed that VCO2 would increase post exercise. Based on PMID: 26213682, I don't see lowering of VCo2. If anything, post exercise, there is a decline in RER. Is this assumption correct?

Author Response

I am sorry but I am not trying to harass the authors. Just wanted to make sure that the correct message is conveyed. 

The authors have assumed that VCO2 would increase post exercise. Based on PMID: 26213682, I don't see lowering of VCo2. If anything, post exercise, there is a decline in RER. Is this assumption correct?

We thank the reviewer for their comment and now better understand their position, which we apologize for not interpreting correctly in our initial revision. The reviewer is indeed correct that RER decreases in the cited paper at the same time VO2 increases. We have clarified the methods to indicate that we assumed VCO2 increases based on the increased resting energy expenditure (which we use for our VCO2 estimate) but acknowledge now in the discussion that VCO2 may have been unchanged, based on the RER (which we cannot confidently conclude with our remote trial and predictive VCO2). Therefore, in light of the reviewer’s helpful comment (and respectful insistence), we have revised the discussion with this possibility (lines 355-360).

Reviewer 2 Report

I have carefully reviewed the revised manuscript. I think the authors have done a good job in addressing the comments and questions raised by me. However, I still insist on description and modifying the manuscript. The comments in this round exist below. Additionally, comments from the previous round that were addressed were deleted, in order to avoid confusion.

Major comment

1)                  The exogenous leucine retention was calculated as the difference between total exogenous leucine oxidation and leucine ingestion. Why did EAA+ and BCAA increase the exogenous leucine retention, although the exogenous leucine oxidation in EAA+ and BCAA groups increased? Its calculation is correct?

Thank you for pointing this out as it may seem counterintuitive. However, the calculations are correct as it is well-documented (via intravenous and oral leucine tracers) that exogenous leucine oxidation increases with the consumption of leucine (PMID: 8997229) and that with foods or beverages higher in leucine elicited greater rates of leucine oxidation (PMID: 19056590). This would be consistent with EAA+ and BCAA increasing exogenous leucine oxidation compared to CHO in the present study.

                          The fraction of exogenous leucine oxidized is commonly around 30-40% (PMID: 8997229), and the remaining 60-70% of exogenous leucine is retained. Therefore, a proportion of the leucine in the EAA+ and BCAA supplements was retained in our study (~200μmol/kg, as shown in Figure 2C). This means that the absolute amount of exogenous leucine retained was greater with EAA+ and BCAA than CHO (which contained only trace amounts of isotopic leucine). Essentially, relative to CHO, both exogenous leucine oxidation and retention was greater with EAA+ and BCAA.

Comments in round 2; Thank you for your description. I well understood amino acid oxidation and retention after amino acids ingestion. I am sorry for my question sentence in previous round. Why were retentions in EAA+ and BCAA large than that in CHO group? The calculus equation is the retention = ingestion – oxidation. Mathematically, does a large oxidation decrease a retention?

2)                  The authors attempted to reveal the effect of ingesting EAAs on exogenous leucine retention as protein synthesis because protein synthesis needs all EAAs as substrate. However, the EAA+ has not only EAAs but other compounds affecting muscle protein metabolism. Thus, it is unclear that EAA in this study caused the results in EAA+.

                          The reviewer is correct in raising this point. We are aware that the presence of glutamine, citrulline, and botanical compounds in the EAA+ supplement may have influenced our findings beyond any effects of the EAAs alone. We discussed the potential role of these compounds in both the Methods and Discussion sections, and stated that these compounds are unlikely to have acute (5h) effects based on preclinical data. This suggests that our findings are likely due to the EAAs in the EAA+ supplement. Additionally, our findings are consistent with studies using supplements only containing EAAs and other amino acids.

Nevertheless, we have now included additional text in the Discussion section to further state that we cannot confirm that these additional compounds in EAA+ did not impact our findings (lines 391-392).

Comments in round 2; Thank you for your description. If ''acute (5h) effects'' are anabolic effects, I recommend the addition of this description for supporting the author’s consideration, with the reference. However, the sentence (these results suggest that a full complement of EAAs is necessary to maximize the anabolic effect of RE. (line 397-398)) cannot be concluded in current results. The finding in current study can conclude that ''EAA formula'' has the anabolic effect.

3)                  The authors used resistance exercise and carbohydrate group (RE+CHO) as control, and compared the effect of EAA+ and BCAA to RE+CHO. However, the CHO did not have nitrogen comparable to EAA+ and BCAA supplements. Nitrogen is critical for amino acid biosynthesis; therefore, it is needed to meet the same nitrogen content as possible.

This is a great point, as many studies investigating amino acid supplements use a non-essential amino acid (NEAA) supplement as an isocaloric control. However, use of an isocaloric CHO control is not without precedence (PMID: 32349353). Both isocaloric NEAA and CHO controls are effective as they eliminate the concern of calories influencing outcomes and do not contain the key compound(s) under investigation (i.e., EAAs/leucine). To our knowledge, there is a lack of research indicating that the use of a NEAA control would impact the outcomes in our study differently than CHO. Nevertheless, our goal was to test the efficacy of essential amino acids in supporting whole body anabolism and made the decision to provide a nominal energy-matched control rather than just water. Regardless, we do not believe our results would be different if we had used a NEAA control condition as they have long been established to have no discernable effect on muscle (PMID: 10198297, 12217881) or whole body protein anabolism (PMID: 30145710).

Comments in round 2; There may be no effect of NEAA on the outcome in the current study. However, even under this study condition, the authors must demonstrate the no effect of nitrogen (e.g. NEAA) on the measurements for suggesting that ''complement of EAAs is necessary to maximize the anabolic effect of RE.'' (line 398). Additionally, the authors cannot reveal the effect of EAA on whole body anabolism in this research design. The research design can answer the efficacy of EAA+ and BCAA for protein metabolism. However, this research design cannot define whether EAA and BCAA itself are needed to increase whole body anabolism as EAA+ contains other compounds and CHO control does not contain nitrogen.

4)                  Due to the COVID-19 pandemic and restrictions to face-to-face research, the authors calculated the exogenous leucine oxidation using an estimated resting VCO2 in described in reference [48]. It is needed to indicate that the method using an estimated resting VCO2 is correct and appropriate.

Thank you for the comment, as the correct calculation of exogenous leucine oxidation is indeed important to our study findings. The calculations used to estimate resting VCO2 in our study (e.g., 300 mmol CO2/hr multiplied by body surface area) are correct and have been used previously in a similar study (PMID: 31765432). In fact, this approach is commonly used in studies involving 13C breath tests to estimate resting VCO2 (PMID: 20443742).

Additionally, recent published work from our lab (PMID: 35609328) indicates that resting VCO2 measured by indirect calorimetry and estimated by the aforementioned calculations yields similar findings. This demonstrates that indirect calorimetry is not essential for estimating exogenous leucine oxidation and an estimation of resting VCO2, as described in the Methods section of our manuscript, is sufficient (we have now included a sentence in the Methods section to state this in lines 204-205).

Comments in round 2; I think that this comment was described and the manuscript was modified. However, I could not confirm the reference (PMID: 35609328). Please send me the reference.
